# Antitumor Activity of Metformin Combined with Locoregional Therapy for Liver Cancer: Evidence and Future Directions

**DOI:** 10.3390/cancers15184538

**Published:** 2023-09-13

**Authors:** Eshani J. Choksi, Mohammad Elsayed, Nima Kokabi

**Affiliations:** 1School of Osteopathic Medicine, Rowan University, Stratford, NJ 08084, USA; choksi47@rowan.edu; 2Interventional Radiology Service, Department of Radiology, Memorial Sloan Kettering Cancer Center, New York, NY 10065, USA; 3Department of Radiology, Division of Interventional Radiology, School of Medicine, University of North Carolina, Chapel Hill, NC 27599, USA

**Keywords:** metformin, hepatocellular carcinoma, mTOR inhibitors, adjuvant therapy

## Abstract

**Simple Summary:**

Existing therapies for hepatocellular carcinoma, although effective, often have limitations in their long-term efficacy. Metformin is increasingly being recognized for its anti-cancer properties and potential in augmenting the tumor response following locoregional therapy. This review analyzed existing comparative studies on combination therapy with metformin and found that metformin had the potential to augment the effects of current locoregional therapies for HCC management.

**Abstract:**

This article aimed to examine the effect of metformin use on improving outcomes after liver-directed therapy in patients with HCC and identify future directions with the adjuvant use of and potential therapeutic agents that operate on similar mechanistic pathways. Databases were queried to identify pertinent articles on metformin’s use as an anti-cancer agent in HCC. Eleven studies were included, with five pre-clinical and six clinical studies. The mean overall survival (OS) and progression-free survival were both higher in the locoregional therapy (LRT) + metformin-treated groups. The outcome variables, including local tumor recurrence rate, reduction in HCC tumor growth and size, tumor growth, proliferation, migration and invasion of HCC cells, HCC cell apoptosis, DNA damage, and cell cycle arrest, showed favorable outcomes in the LRT + metformin-treated groups compared with LRT alone. This systemic review provides a strong signal that metformin use can improve the tumor response after locoregional therapy. Well-controlled prospective trials will be needed to elucidate the potential antitumor effects of metformin and other mTOR inhibitors.

## 1. Introduction

Since 1980, the incidence of liver cancer has more than tripled and death rates have doubled, indicating a clear need for aggressive diagnostic and treatment modalities to reduce the burden of this disease in the United States and worldwide. Hepatocellular carcinoma (HCC) is classified as the most common type of liver cancer, accounting for approximately 80% of all diagnoses [1]. It also is a leading cause of cancer mortality worldwide, with a 5-year survival of under 20% [2,3,4]. Although studies show a plateau or even a decrease in the number of new HCC cases in recent years [2,3], previous Surveillance, Epidemiology, and End Results (SEER) database analyses have projected that there will still be over 56,000 cases of HCC by 2030 in the US and 22 million cases in the next two decades worldwide [4,5]. The most common risk factors associated with the development of this malignancy include viral hepatitis, primarily Hepatitis B and C, alcoholic liver disease and cirrhosis, and, more recently, non-alcoholic fatty liver disease [6,7]. Several studies have also identified a correlation between metabolic syndrome, male gender, and Hispanic and African Americans with higher incidences of hepatocellular carcinoma diagnosis [8].

The plateau in hepatocellular carcinoma cases in recent years is likely attributed to more aggressive viral hepatitis prevention measures, rather than successful and efficient treatment interventions for HCC [9], which highlights that, although there may be implemented strategies to reduce the number of new HCC cases, the staggering mortality rate shows that there is still limited treatment success in patients who are suffering the consequences of HCC [10]. The 2-year survival rates in the United States have remained under 50%, and 5-year survival rates of HCC remain below 12% [11,12]. 

There are various treatment options for HCC, including liver transplantation, curative resection, radiofrequency ablation (RFA), chemoembolization, and systemic targeted agents. The treatment of the cancer depends on the stage of the tumor at diagnosis, patient performance status, and liver function, requiring a holistic evaluation of the patient before deciding on the best treatment modality. Surgical resection and liver transplantation offer the best curative treatments for HCC [13]; however, eligibility for these procedures requires detection at an early stage, and, unfortunately, the majority of patients tend to be diagnosed at advanced disease stages [2,14,15,16,17,18,19,20]. In patients with more advanced disease, other treatments, such as radiofrequency ablation (RFA), transarterial chemoembolization (TACE), and yttrium 90 (Y90) radioembolization, have shown to be successful at reducing tumor progression, with radioembolization having the safest toxicity profile [20,21,22,23,24]. Some studies have shown that these treatments are equally effective in achieving tumor necrosis [25,26,27], while others identified that Y90 radioembolization leads to significantly higher survival rates in comparison with transarterial chemoembolization [28,29]. Despite recent advances in systemic and locoregional therapies, prognosis for many patients with unresectable HCC remains poor due to limited responses and a high rate of recurrence. The global disease burden and poor outcomes associated with HCC have prompted efforts to identify strategies to improve responses to existing therapies, including the use of adjunctive agents with liver-directed therapies (LDTs) [30,31,32,33,34,35]. 

Researchers have reported improved outcomes associated with the use of various adjuvant drugs combined with locoregional therapies, such as longer overall survival and delayed HCC tumor recurrence when LRT was combined with angiotensin II receptor 1 blockers (sartans) [36] and angiotensin-converting enzyme inhibitors (ACE-I) [37,38,39]. Additionally, adjuvant use of antiviral agents with LRT has been of interest for decades, as viral hepatitis is a primary driver of a significant proportion of patients with HCC, and the use of such drugs has been associated with improved overall and recurrence-free survival [40,41,42,43,44,45,46,47,48]. Vitamin K analogs and retinoids have also shown promise as an adjuvant therapy against HCC, given various anti-tumor effects, including the induction of cell cycle arrest and inhibition of HCC cell proliferation [38,49,50,51,52,53,54,55]. 

The interest in understanding the role of metformin in cancer prevention was first sparked by an observational study conducted in 2005 that showed its ability to suppress the incidence and progression of various cancers through several mechanisms, including the inhibition of insulin-like growth factor-1 (IGF-1) [56], activation of the adenosine 5′-monophosphate-activated protein kinase (AMPK) [57], inhibition of the mammalian target of rapamycin complex I (mTORC1) [58], inhibition of nuclear factor-kB (NF-kB) [59], and inhibition of oxidation phosphorylation in cancer cells [60,61]. Subsequent studies have shown a protective effect of metformin in preventing the onset of liver cancer, specifically hepatocellular carcinoma [62,63,64,65,66,67,68,69,70]; however, further investigation is warranted to identify the true capacity of metformin in the setting of hepatocellular carcinoma. A growing number of population studies and meta-analyses have found that metformin use is associated with decreased incidence and mortality for various malignancies, including prostate, endometrial, gastric, pancreatic, thyroid, breast cancer, and colon cancer [71,72,73,74,75]. Recent preclinical and clinical studies indicate that metformin use plays a chemo-preventative role in HCC development through mTOR inhibition [69,76,77,78]. Additionally, the utilization of stereotactic body radiation therapy (SBRT) has shown promising results in cases of inoperable HCC [79,80], highlighting the potential of external-beam radiotherapies as a novel locoregional therapy that can also be combined with adjunctive agents.

Metformin use shows particular promise in HCC, with large studies across various geographic locations and clinical settings showing that its use is associated with reduced incidence of HCC in patients with cirrhosis, as well as improved prognosis among patients with HCC. This systematic review compiles current literature regarding the effect of metformin use on improving outcomes after liver-directed therapy in patients with HCC. Future directions regarding the adjuvant use of metformin and potential therapeutic agents that work on similar mechanistic pathways will be discussed.

## 2. Materials and Methods

A systematic review was performed on the existing literature examining metformin as an adjuvant agent in patients with hepatocellular carcinoma undergoing locoregional therapy. All of the literature search findings were reported in accordance with the Preferred Reporting Items for Systematic Review and Meta-Analysis (PRISMA) guidelines. Since no human subjects were studied in this research, this systematic review was exempt from official Institutional Review Board (IRB) approval. All studies were uploaded to EndNote and were screened using the Covidence systematic review software (Veritas Health Innovation, Melbourne, Australia, www.covidence.org. URL accessed on 27 December 2022). This review was not registered in PROSPERO.

### 2.1. Literature Search and Strategy

The PubMed, Cochrane, EMBASE, and MEDLINE library databases were each queried in December 2022. Each database was searched for any relevant studies pertaining to metformin’s use as an anti-cancer agent in HCC. Articles published between 1981 and 2022 were analyzed. The search terms for this systematic review contained the following combinations of terms: “metformin”, “hepatocellular carcinoma”, “drug response”, “type 2 diabetes mellitus”, “tumor sensitizing”, “locoregional therapy”, “transarterial chemoembolization”, “transarterial radioembolization”, “radiofrequency thermal ablation”, “yttrium-90 radioembolization”, and “external beam radiation therapy”. The reference sections of each eligible full-text article were reviewed to ensure that additional articles were not missed during the initial database literature search.

### 2.2. Eligibility and Search Criteria

Articles were selected by first reviewing the abstracts for relevance, followed by full-text assessment according to the stated inclusion and exclusion criteria. All duplicate articles were removed in Covidence during the initial screening process.

The eligibility criteria for this systematic review included the following: (1) the study enrolled adult patients (>18 years); (2) the patients in the study had a diagnosis of hepatocellular carcinoma; (3) the study consisted of patients who underwent locoregional therapy for HCC, including radiofrequency thermal ablation (RFA), yttrium-90 (Y90) radioembolization, transarterial chemoembolization (TACE), transarterial radioembolization (TARE), microwave ablation (MWA), and external-beam radiotherapy (EBRT), including conventional external beam radiation, stereotactic body radiotherapy (SBRT), hypofractioned radiotherapy (HypoRT), and particle therapy (gamma ray or neutron ray); (4) patients were concomitantly taking metformin during LDT. Any studies that were (1) case series including fewer than four patients, (2) case reports including fewer than four patients, (3) reviews, letters, editorials, or animal studies, or (4) did not report relevant outcomes were excluded during the initial screening of titles and abstracts. Studies that reported outcomes on patients undergoing liver resection, transplantation, or systemic chemotherapies were also excluded.

### 2.3. Data Extraction

Data extraction was performed on all articles that were deemed eligible following full-text assessment. A custom table was generated in Microsoft Excel (version 2019; Microsoft, Redmond, VA, USA) to organize all contents of the data extraction. The following information was collected through data extraction: title, author, publication year, location of the study performed, number of patients, male-to-female ratio, age of patients, control and experimental groups, human cell type (if applicable), metformin dosage, study outcomes, overall survival, and progression-free survival.

### 2.4. Statistical Analysis

Aggregate data of all outcomes were collected, and the average values were calculated using the weighted average mean approach. If there was an instance where only the median and interquartile range of an outcome variable was provided, the average was calculated using Hozo’s formula [81]. All standard deviation calculations were performed using Hozo’s pooled standard deviation formula [81].

## 3. Results

### 3.1. Study Selection

There were 340 studies initially identified through the database search. Following title and abstract screening, there were 154 studies deemed eligible for full-text assessment. Eleven studies were ultimately included in the systematic review [82,83,84,85,86,87,88,89,90,91]. One study included both pre-clinical and clinical data and was thus considered two separate studies [89]. Figure 1 shows a PRISMA diagram outlining the selection process for eligible studies and reasons for exclusion.

### 3.2. Study and Patient Characteristics

Both pre-clinical and clinical studies were identified through the literature search process. Of the 11 total studies included in this systematic review, 5 studies were classified as human preclinical studies [87,88,89,90,91] and 6 studies were classified as human clinical studies [82,83,84,85,86,89]. All included studies were full-text publications and were published between 2012 and 2022. Among the 11 studies, 2 studies performed radiofrequency ablation (RFA) [82,90], 3 studies performed transarterial chemoembolization (TACE) [83,86,89], and 1 study performed Y90 radioembolization (Y90 RS) [84]. Four of the eleven studies performed external beam radiation therapies, which included stereotactic body radiotherapy (SBRT)/hypofractionated radiotherapy (HypoRT) (one study) [85], gamma-ray radiation/neutron radiation (two studies) [87,91], and ionizing radiation (one study) [88]. There was a total of 805 patients among the six clinical studies, with males representing 50.6% of total patients. A total of 581 patients were in the control group (locoregional therapy only) in the studies and 233 patients were in the experimental group (locoregional therapy + metformin combination therapy) in the studies. The mean age of patients was 65.6 ± 9.7 years (Table 1).

### 3.3. Effect of Metformin and Locoregional Therapy on Survival in Patients

There were four clinical studies reporting outcomes related to overall survival [82,83,85,89] and four clinical studies reporting outcomes on progression-free survival [82,83,85,86] (Table 2). Overall survival ranged from 12 months to 75 months, with a mean overall survival of 39.8 months in the locoregional therapy-treated group alone, and overall survival ranged from 12 months to 79 months with a mean overall survival of 39.1 months in the locoregional therapy + metformin-treated group. (Table 2). The progression-free survival ranged from 7 months to 74 months, with a mean progression-free survival of 35.5 months in the locoregional therapy-treated group alone, and progression-free survival ranged from 12 months to 79 months with a mean progression-free survival of 37.2 months in the locoregional therapy + metformin-treated group. (Table 2). Metformin + locoregional therapy resulted in improved overall survival (*p* < 0.001) and progression-free survival (*p* < 0.001) compared with locoregional therapy alone.

### 3.4. Tumor Recurrence

Two clinical studies analyzed tumor recurrence after either TACE alone or TACE + metformin combination therapy [83,86]. Both clinical studies reported lower local tumor recurrence in TACE + metformin-treated patients compared with TACE monotherapy (Table 2). Jung et al. 2022 reported local tumor recurrence of 85.2% in non-metformin-treated patients and 74.8% in metformin-treated groups. Chen 2022 reported that tumor recurrence for non-metformin- and metformin-treated groups was 52.39% vs. 29.19% at 1 year, 17.03% vs. 0.00% at 2 years, and 4.26% vs. 0.00% at 3 years.

### 3.5. Tumor Growth and Size

One clinical study (Table 2) [84] and two pre-clinical studies (Table 3) [34,90] reported outcomes related to tumor size and growth. All studies reported that combination therapy resulted in a greater reduction in HCC tumor growth and size in comparison with monotherapy. Similarly, Elsayed et al. 2021 reported changes in tumor growth as a change in total tumor diameter (TTD), revealing metformin + TACE was associated with a greater mean TTD reduction of 40.4 mm, compared with 17.1 mm in patients receiving TACE alone. (*p* = 0.018).

### 3.6. Tumor Proliferation, Migration, and Invasion

Two pre-clinical studies reported outcomes related to HCC cell proliferation, migration, and invasion (Table 3) [89,90]. Cell proliferation is defined as the rate of local tumor growth, migration is the directed movement of cells in response to chemotactic signals, and invasion is the vascular invasion of cancerous cells. All studies stated that a combination regimen of metformin use was associated with decreased tumor growth, proliferation, migration, and invasion of HCC cells in comparison with LRT alone.

### 3.7. Tumor Apoptosis, DNA Damage, and Cell Cycle Arrest

Two pre-clinical studies reported outcomes related to HCC cell apoptosis, DNA damage, and cell cycle arrest (Table 3) [89,93]. Both reported increased apoptosis, DNA damage, and cell cycle arrest in metformin combination therapy compared with LRT alone. 

## 4. Discussion

This systematic review includes both preclinical and clinical data suggesting that metformin use may improve outcomes when used in conjunction with locoregional therapies. The preclinical studies analyzed in this review collectively found that metformin use could sensitize tumors to locoregional therapy, specifically by impairing several tumorigenic properties, including cell proliferation and migration. In addition, this systematic review is the first known to include human clinical data on locoregional therapy + metformin, compared with prior review articles that only included studies in animal models [94]. 

The results of our study found multiple preclinical and clinical outcomes supporting the utilization of metformin to augment the tumor response in patients with hepatocellular carcinoma. The six included clinical studies demonstrated superior survival outcomes in metformin plus locoregional therapy compared with locoregional monotherapy alone. We found that both the mean overall survival and mean progression-free survival were statistically higher in combination therapy-treated patients. When analyzing the effects of combination therapy on tumor recurrence, statistical analysis confirmed that the combination therapy with TACE and metformin resulted in statistically lower tumor recurrence rates compared with TACE monotherapy alone. Similarly, the comparison of tumor recurrence at 1 year, 2 years, and 3 years showed lower tumor recurrence percentages in metformin + locoregional therapy combination groups compared with locoregional therapy alone. Additional studies with various locoregional therapy may further support these results, as there were no identified studies that analyzed additional LRT + metformin specifically on tumor recurrence rates. All studies that compared variables correlating with tumor size and growth showed that combination therapy with metformin and locoregional therapy resulted in less tumor growth and smaller tumor sizes compared with patients treated with locoregional therapy alone. Preclinical studies reported data on the remaining variables, including tumor proliferation, migration, invasion, apoptosis, DNA damage, and cell cycle arrest. All preclinical studies reporting data on these variables identified superior outcomes with combination therapy in comparison with monotherapy alone. The combination of these results shows superior outcomes with metformin combination therapy when studying a variety of markers correlating to tumor growth progression. 

Interestingly, there are several published literature reviews on the synergistic effects of metformin in treating other types of cancers. One published review found that, in patients with early stage colorectal cancer, metformin use was associated with significant benefits in recurrence-free survival, overall survival, and cancer-specific survival [95]. Additionally, this review found that, in patients with early stage prostate cancer, metformin was also associated with significant benefits in recurrence-free survival, overall survival, and cancer-specific survival [95]. This review, however, found that there were no significant benefits of metformin in treatment for breast and urothelial cancer. Another meta-analysis specifically analyzing the role of metformin in colorectal cancer management showed that patients already taking metformin had a significantly lower incidence of colorectal cancer. This review also determined that overall survival and cancer-specific survival were significantly higher in patients taking metformin compared with those who were not [96]. One study interestingly performed a study to estimate the overall cancer risk of type 2 diabetic patients already taking metformin. The study showed that patients taking metformin had a significantly lower risk of developing cancer, highlighting that metformin may be an independent protective factor for cancer risk entirely [97]. Another systematic review combined papers that, overall, supported the notion that metformin has clinical benefits in the treatment of gynecologic cancers, specifically cervical cancer [98]. Despite the results of prior systematic reviews, there is emerging literature on combination therapy with PD-1 inhibition plus rapamycin and metformin, which has been shown to enhance the anti-tumor efficacy in triple-negative breast cancer, highlighting promising therapies in years to come [99]. The ability of metformin to augment therapies in several cancer types highlights the versatility of the drug, as well as the legitimacy of its therapeutic benefits. 

Although not fully understood, it is hypothesized that the major driving mechanism of metformin’s anti-cancer activity is likely related to the inhibition of the mammalian target of rapamycin (mTOR) pathway. There are two complexes within the mTOR pathway, mTORC1 and mTORC2, which both contain a catalytic subunit known as mTOR. The two complexes are known to receive stimuli from various hormonal signals throughout the body and are known to be imperative for cellular growth processes. mTORC1 primarily receives signals from pathways, including insulin-like growth factor 1 (IGF-1), insulin-like growth factor 2 (IGF-2), and AMP-activated protein kinase (AMPK). The mTOR pathway plays a key role in nutrient-sensing and regulation of various cellular processes, including cell proliferation, DNA repair, and protein synthesis. Metformin is found to specifically inhibit mTORC1 via the AMPK pathway through the activation of the tumor suppressor genes TSC1 and TSC2, which code for the tuberin and hamartin proteins, respectively. Additionally, metformin inhibits mTORC1 directly via AMPK with the aid of AMPK-inhibiting RAPTOR (regulatory-associated protein of mTOR) [100]. AMPK phosphorylates RAPTOR, and this phosphorylation is required for the inhibition of the mTORC1 complex. RAPTOR is an adaptor protein and positive regulator within the mTORC1 complex. The overactivation of mTOR is also associated with the development of various cancers, thus creating an opportunity to target this pathway with metformin and other therapeutic agents [101,102]. Metformin has also been shown to induce p53, which is a tumor-suppressor protein with the ability to also inhibit mTORC1. p53 is able to detect DNA damage and other genotoxic stresses that change the genetic material of cells. As a result, p53 is stimulated so that cellular growth and proliferation are halted to prevent the progression of abnormal cell growth. p53 also activates AMPK, and subsequently TSC1 and TSC2, which then inhibit mTORC1 [103]. Finally, metformin has also been found to reduce the production of fatty acid synthase, which is known to be oncogenic and significantly upregulated in cancer cells. Metformin studies have also found that it is able to reduce the production of reactive oxygen species, thus reducing oxidative stress and DNA damage through the inhibition of mitochondrial complexes [104].

The major role of mTOR in tumorigenesis has prompted the further investigation of potentially more potent mTOR inhibitors. Emerging literature suggests that the mTOR inhibitors anthramycin and Torin-2, for example, may be effective in treating HCC and suppressing liver cancer stem cells [45,105]. The efficacy of these agents may be further improved by loading them in specialized delivery systems, such as nanoparticle beads. The development of nanoparticles to deliver metformin to cancer cells has become of increasing interest, as the particles have the potential to successfully deliver the drug directly to the cancer cell of interest without systemic absorption or adverse side effects. A study conducted by Snima et al. analyzed the ability of O-carboxymethyl chitosan nanoparticles for the delivery of metformin to pancreatic cancer cells [106]. The study reported that cytotoxicity studies showed that the drug-incorporated nanoparticles induced statistically significant toxicity in the pancreatic cancer cells compared with normal healthy cell lines. Additionally, metformin-loaded nano-particles successfully demonstrated non-specific internalization by both normal and pancreatic cancer cells; however, the metformin released by nanoparticles preferentially induced toxicity in the pancreatic cancer cells, rather than the healthy cells. 

This systematic review had several inherent limitations, primarily related to the methodology of the included studies. The included studies had limited data regarding the stage, extent of disease burden, and tumor biology, which precludes the ability to perform certain sub-analyses and further elucidate the effect of metformin. Similarly, there is likely significant heterogeneity regarding patient selection and how the locoregional therapy was technically performed, so the lack of standardization may have influenced the analysis in this study. Another limitation is that older studies may not have utilized contemporary treatment approaches, such as modern dosing strategies using Y90 radioembolization, which may have resulted in worse outcomes and obscured the full therapeutic effect of metformin. This systematic review is also limited by the fact that metformin is typically prescribed to patients with diabetes mellitus, which may introduce multiple layers of confounding bias. Additional studies on patients who do not have a concomitant diagnosis of diabetes mellitus may help to further understand the true efficacy of metformin as an adjuvant agent. Finally, the usage of metformin is rarely associated with life-threatening hepatotoxicity; therefore, patients being treated with metformin in the setting of hepatocellular carcinoma should be closely monitored for signs of acute liver failure. The included studies did not report any cases of life-threatening hepatotoxicity associated with metformin use; however, prospective trials would be useful to elucidate potential hepatotoxicity in this study.

Finally, this systemic review does not address the potential interactions between metformin and the recently approved checkpoint inhibitors. Emerging evidence suggests that metformin may augment the effect of checkpoint inhibitors [107]. Therefore, studies investigating the combination of these agents with locoregional therapy warrant further investigation. 

## 5. Conclusions

Despite the limitations of this study, this systemic review provides a strong indication that metformin use can improve the tumor response after locoregional therapy. We found robust outcomes following combination therapy with metformin and locoregional therapy, including improved overall survival, reduced tumor recurrence, decreased cell proliferation and migration, and increased cancer cell DNA damage and apoptosis. Well-controlled prospective trials will be needed to elucidate the full potential antitumor effects of metformin and other mTOR inhibitors. Additional studies may help to fully understand the survival advantage suggested by this study.

## Figures and Tables

**Figure 1 cancers-15-04538-f001:**
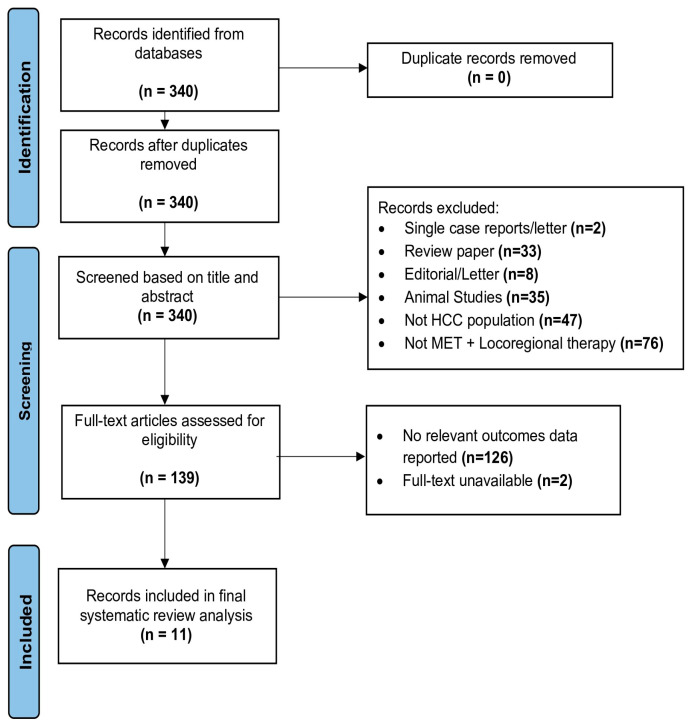
Preferred Reporting Items for Systematic Reviews and Meta-analyses (PRISMA 2020) diagram.

**Table 1 cancers-15-04538-t001:** Demographics of included preclinical and clinical studies.

Author, Year	Study Type	Location of Study	No. of Patients ^1^	Sex, M:F Ratio	Mean Age (Years)	Type of HCC Intervention ^2^	Control Group (# of Patients) ^3^	Experimental Group (# of Patients) ^3^
Chen 2011 [82]	Human clinical	Taiwan	135	84:51	63.0 ± 9.7	LDT	RFA (114)	RFA + MET (21)
Chen 2022 [83]	Human clinical	China	123	108:15	-	LDT	TACE (73)	TACE + MET (50)
Elsayed 2021 [84]	Human clinical	USA	103	72:34	64.2 ± 8.9	LDT	Y-90 RS (93)	Y-90 RS + MET (19)
Jang 2015 [85]	Human clinical	Korea	217	57:160	-	EBRT	SBRT or HypoRT (198)	SBRT or HypoRT + MET (10)
Jung 2022 [86]	Human, clinical	South Korea	164	75:89	69.0 ± 16.4	LDT	TACE (73)	TACE + MET (91)
Kim 2014 [87]	Human pre-clinical	South Korea	-	-	-	EBRT	y-ray radiation and neutron radiation	y-ray radiation/neutron radiation + MET
Liu 2012 [88]	Human pre-clinical	China	-	-	-	EBRT	Ionizing radiation	Ionizing radiation + MET
Tian 2016 [89]	Human pre-clinical/clinical	China	63	11:52	-	LDT	TACE (30)	TACE + MET (33)
Zhang 2017 [90]	Human pre-clinical	China	-	-	-	LDT	RFA	RFA + MET
Zhang 2019 [91]	Human pre-clinical	China	-	-	-	EBRT	Gamma ray irradiation	Gamma ray irradiation + MET

^1^ Pre-clinical studies did not have patients; ^2^ LDT = liver-directed therapy, EBR = external-beam radiation therapy; ^3^ TACE = transarterial chemoembolization, MET = metformin, RFA = radiofrequency ablation, RS = radiosensitization, SBRT = stereotactic body radiotherapy, HypoRT = hypofractioned radiotherapy.

**Table 2 cancers-15-04538-t002:** Outcomes of human clinical studies in control and experimental groups.

Study	Overall Survival (Control)	Overall Survival (Experimental)	Progression-Free Survival (Control)	Progression Free Survival (Experimental)	Results of Study ^1^	Metformin Dosage
Jung 2022 [86]	-	-	74 months	79 months	Metformin use associated with higher ORR, lower LTR, better IR, and lower recurrence	-
Elsayed 2021 [84]	39.7 months	33.7 months	-	-	MET + Y-90 RS had greater reduction in TTD, number of lesions, and % TTD	-
Jang 2015 [85]	37% 2 years	76% 2 years	16% 2 years	46% 2 years	Use of metformin in patients receiving radiotherapy was associated with higher OS	1000 mg/day
Chen 2011 [82]	93.9% 1 year; 80.2% 3 years; 64.7% 5 years	95.0% 1 year; 69.2% 3 years; 60.5% 5 years	74.5% 1 year; 44.8% 3 years; 26.2% 5 years	95% 1 year; 69.2% 3 years; 60.5% 5 years	Metformin users among diabetic patients with HCC undergoing RFA had favorable overall survival compared with patients without metformin	750 mg/day
Chen 2022 [83]	32 months	42 months	7 months	12 months	Metformin increased long-term survival time and significantly prevented recurrence after TACE	2 g/day
Tian 2016 [89]	75 months	79 months	-	-	Metformin + TACE combination resulted in increased OS	-

^1^ ORR = overall response rate, LTR = local tumor recurrence, IR = initial response, TTD = total tumor diameter, RFA = radiofrequency ablation, HCC = hepatocellular carcinoma, TACE = transarterial chemoembolization, OS = overall survival, PFS = progression-free survival.

**Table 3 cancers-15-04538-t003:** Outcomes of Human Pre-Clinical Studies in Control and Experimental Groups.

Study	Cell Type Used	Results of Study ^1^	Metformin Dosage
Zhang 2017 [90]	HepG2 and SMMC7721	MET + RFA inhibited proliferation, migration, and invasion of Hep G2 cells after insufficient RFA. MET also blocked growth of HepG2 cells	0.5 mg/mL
Zhang 2019 [91]	SMMC-7721	Metformin demonstrated enhanced radiosensitivity and inhibition of EMT in HCC cells.	0, 1, 5, 10, and 20 mM
Tian 2016 [89]	Bel7402 and Bel7402-5 fluorouracil	5-FU + MET increased chemotherapeutic sensitivity, decreased cell proliferation, increased apoptosis, and arrested cell cycle at G0/G1	2 mg/kg
Liu 2012 [88]	HepG2 and Bel-7402 cells	Combination of Met + ionizing radiation inhibited cell survival, cell cycle progression, and increased DNA damage more than monotherapies	-
Kim 2014 [87]	Huh7, HepG2 and Hep3B	Combination of y-ray radiation + MET enhanced radiation efficacy, increased apoptosis, increased DNA damage, and stopped G2/M cell cycle in HCC cells	-
Xin 2016 [92]	PLC/PRF/5, SK-Hep-1, Huh7	MET increased effects of Sorafenib and significantly decreased viability/proliferation of HCC cells	200 μM

^1^ MET = metformin, HCC = hepatocellular carcinoma, RFA = radiofrequency ablation, EMT = epithelial-to-mesenchymal transition, 5-FU = 5-fluorouracil.

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
