# Peer review of "Antitumor Activity of Metformin Combined with Locoregional Therapy for Liver Cancer: Evidence and Future Directions"

_cancers, 2023, doi:10.3390/cancers15184538_

Round 1
Reviewer 1 Report (New Reviewer)
In this research, the authors reviewed the status of antitumor activity of metformin combined with locoregional therapy for liver cancer: evidence and future directions. In my opinion, the current version of this manuscript fits the scope of Cancers and could be accepted after major revision.
My specific comments are in detail listed below:
1. In Line 72-85, the current development of Metformin and its derivative as immune regulating agent should be revealed and introduced. Some references should be added to this part including 10.1016/j.jconrel.2022.11.004.
2. Some minor mistakes exist in the references. The authors should correct it.
3. In Line 223-226, how metformin affect the DNA repair process should be added and clearly discussed. Some references should be added to this part including 10.1016/j.ijbiomac.2022.10.167.
4. Some references are out of date. Some new recent references may be better.
5. In conclusion, the clinical transformation barriers of metformin combined with locoregional therapy for liver cancer should be better out-looked.
Author Response
Please see the attachment.

Reviewer 2 Report (New Reviewer)
Very interesting and well written study. The authors should try to pool some estimates.
THe authors should comment on the general state of art of the combination of locoregional therapies and other drugs, for example sartans (cite PMID: 25974743)
The authors should comment on the potential hepatotoxicity of metformin
The authors should comment on the general state of art on the influence of diabetes on HCC incidence and clinical course (cite PMID: 23845075 )
Round 2
Reviewer 2 Report (New Reviewer)
The revised version of the paper is OK. Thank you!
This manuscript is a resubmission of an earlier submission. The following is a list of the peer review reports and author responses from that submission.
Round 1
Reviewer 1 Report
The authors purpose a systematic review on the topic. Of the 11 total studies included in this systematic review, 5 studies were classified as human preclinical studies and 6 studies were classified as human clinical studies. All included studies were full-text publications and were published between 2012 to 2022, in journal with low impact factor and low editorial requirements. Among the 11 studies, the interventional procedures are not the same and not comparable : 2 studies performed radiofrequency ablation (RFA), 3 studies performed transarterial chemoembolization (TACE), and 1 performed Y90 radioembolization (Y90 RS). Four of the 11 studies performed external beam radiation therapies, which included stereotactic body radiotherapy (SBRT)/hypofractionated radiotherapy (HypoRT) (1 study), gamma-ray radiation/neutron radiation (2 studies), and ionizing radiation (1 study). For the 6 clinical studies, the patient cohort is small enough to allow for robust statistical conclusions.
Survival data are only available for 4 studies, and it's hard to understand how the authors arrived at methodologically reliable statistical conclusions (in favor of adding metformin) from published data (mean OS 39.1 months vs. 39.8 months (p<0.001) and mean PFS 37.2 months vs. 35.5 months (p<0.001) for local intervention +metformin vs. local intervention alone respectively).
The 2-4 month lag between PFS(mean) and OS(mean) is also very surprising at this stage of patient management (BCLCCA or B).
The rest of the results are not uninteresting (even if very fragmentary), but do not provide convincing evidence.
The introduction and conclusion are rather well-executed, but then again the subject of metformin in oncology has already been covered many times over in numerous publications, and this doesn't bring any specific originality.
Reviewer 2 Report
Metformin is increasingly recognized for its anti-cancer properties and potential to enhance tumor response after local treatment. This review analyzes the existing comparative studies of Metformin combination therapy, and finds that Metformin may enhance the effect of local treatment of liver cancer. The manuscript is well organized and has prominent viewpoints.
Although the authors have made a Systematic review of the existing literature, they have studied the role of Metformin as an adjuvant in patients with hepatocellular carcinoma receiving local treatment. However, the necessary mechanisms of action should be further refined and presented in graphical form. Because the advantage of doing so is that it is easy for readers to understand and read.
The correlation between hepatitis vaccination and the cure rate of liver cancer treatment should be further discussed.